

# SCelVis: exploratory single cell data analysis on the desktop and in the cloud

Benedikt Obermayer[1,2,*], Manuel Holtgrewe[1,2,*], Mikko Nieminen[1,3], Clemens Messerschmidt[1,2] and Dieter Beule[1,3]

[1] Core Unit Bioinformatics, Berlin Institute of Health, Berlin, Germany
[2] Charité—Universitätsmedizin Berlin, Berlin, Germany
[3] Max Delbrück Center for Molecular Medicine, Berlin, Germany
[*] These authors contributed equally to this work.

## ABSTRACT

**Background**. Single cell omics technologies present unique opportunities for biomedical and life sciences from lab to clinic, but the high dimensional nature of such data poses challenges for computational analysis and interpretation. Furthermore, FAIR data management as well as data privacy and security become crucial when working with clinical data, especially in cross-institutional and translational settings. Existing solutions are either bound to the desktop of one researcher or come with dependencies on vendor-specific technology for cloud storage or user authentication.

**Results**. To facilitate analysis and interpretation of single-cell data by users without bioinformatics expertise, we present SCelVis, a flexible, interactive and user-friendly app for web-based visualization of pre-processed single-cell data. Users can survey multiple interactive visualizations of their single cell expression data and cell annotation, define cell groups by filtering or manual selection and perform differential gene expression, and download raw or processed data for further offline analysis. SCelVis can be run both on the desktop and cloud systems, accepts input from local and various remote sources using standard and open protocols, and allows for hosting data in the cloud and locally. We test and validate our visualization using publicly available scRNA-seq data.

**Methods**. SCelVis is implemented in Python using Dash by Plotly. It is available as a standalone application as a Python package, via Conda/Bioconda and as a Docker image. All components are available as open source under the permissive MIT license and are based on open standards and interfaces, enabling further development and integration with third party pipelines and analysis components. The GitHub repository is https://github.com/bihealth/scelvis.

Corresponding author
Dieter Beule, dieter.beule@bihealth.de

## INTRODUCTION

Single-cell omics technologies, in particular single-cell RNA sequencing (scRNA-seq), allow for the high-throughput profiling of gene expression in thousands to millions of cells with unprecedented resolution. Recent large-scale efforts are underway to catalogue and describe all human cell types (*Regev et al., 2017*) and to study cells and tissues in health and disease (https://lifetime-fetflagship.eu). Single-cell sequencing could therefore

become a routine tool in the clinic for comprehensive assessments of molecular and physiological alterations in diseased organs as well as systemic responses, e.g., of the immune system. The enormous scale and high-dimensional nature of the resulting data presents an ongoing challenge for computational analysis (*Stegle, Teichmann & Marioni, 2015*). Ever more sophisticated methods, e.g., deep learning frameworks (*Eraslan et al., 2019*), extract multiple layers of information from cell types to lineages and differentiation programs. Many of these methods, their mathematical background, and the underlying assumptions will remain opaque to users without specific bioinformatics expertise. At the same time, an in-depth understanding of the relevant biology is often beyond the know-how of typical bioinformatics researchers. More than ever, single-cell omics requires close communication and collaboration from wet and dry lab experts. Due to the large amount of data, communication needs to be based on interactive channels (e.g., web-based apps) rather than static tables. Further, as single-cell omics moves towards the clinic, FAIR (*Wilkinson et al., 2016*) data management, data privacy, and data security issues need to be handled appropriately. All employed methods should be able to scale towards handling a large number of users and even larger numbers of samples.

### State of the art

Web apps have been used extensively in the single-cell literature and are most commonly built on Shiny (*Winston et al., 2019*). Table 1 presents an overview of mostly web-based visualization tools for single-cell data. For instance, Pagoda2 (*Fan et al., 2016*) comes with a simple intuitive web app but is limited to data processed with Pagoda2. Cerebro (*Hillje, Pelicci & Luzi, 2019*) is a Shiny web app and provides relatively rich functionality such as gene set enrichments and quality control statistics, but the input is limited to Seurat objects, similar to the Single Cell Viewer (SCV; *Wang et al., 2019*) which also relies on Shiny. CellexalVR (*Legetth et al., 2018*) provides an immersive virtual reality platform for the visualization and analysis of scRNA-seq data, but requires special hardware and runs only on Windows 10. Cellxgene (https://chanzuckerberg.github.io/cellxgene/) is very fast and user-friendly but restricted to visualizing two-dimensional embeddings. Finally, the Broad Single Cell Portal (https://portals.broadinstitute.org/single_cell) provides a large-scale web service for a large number of users and studies. It includes a 10X Genomics data processing pipeline and user authentication/account management. However, the underlying Docker image strongly depends on vendor-specific cloud systems such as Google Cloud and Broad Firecloud services. Its implementation thus poses practical hurdles, in particular if it is to be integrated into existing clinical infrastructure.

## MATERIALS & METHODS

SCelVis is based on Dash by Plotly (*Plotly Technologies Inc., 2015*) and accepts data in HDF5 format as AnnData objects. These objects can be created using Scanpy (*Wolf, Angerer & Theis, 2018*), provide a scalable and memory-efficient data format for scRNA-seq data and integrate naturally into python environments. SCelVis also provides conversion functionality to AnnData from raw text, loom format or 10X Genomics CellRanger output. The built-in converter is accessible from the command line and a web-based user interface

**Table 1 Comparison of single-cell visualization tools.**

| | Pagoda | Cerebro | Single Cell Viewer | CellexalVR | Cellxgene | Single Cell Portal | ScelVis |
|---|---|---|---|---|---|---|---|
| Reference | *Fan et al. (2016)* | *Hillje, Pelicci & Luzi (2019)* | *Wang et al. (2019)* | *Legetth et al. (2018)* | chanzuckerberg. github.io/ cellxgene/ | singlecell. broadinstitute.org | This study |
| Version | v0.1.1 | v1.1.0 | | v0.13.0 | v0.13.0 | v0.38.0 | v0.8.1 |
| Language | R | R | R | C# | Python | ruby | Python |
| Platform | Browser | Browser | Browser | Windows 10 | Browser | Browser | Browser |
| License | GPL-3.0 | MIT | GPL-3.0 | GPL-3.0 | MIT | BSD 3-clause | MIT |
| Additional dependencies | None | None | None | HTC Vive Controller | None | Google Cloud Platform etc. | None |
| Plot types | Scatter, heatmap | Scatter, violin, heatmap, box, bar | Scatter, heatmap, dot | Scatter, heatmap | Scatter, histogram | Scatter, violin, heatmap, box | Scatter, violin, box, bar, dot |
| Cell filtering | ✓ | ✓ | ✓ | ✓ | ✓ | ✕ | ✓ |
| Differential expression | ✓ | ✓ | ✓ | ✓ | ✓ | ✕ | ✓ |
| Data input | Local + remote | Local | Local | Local | Local + remote | Local + remote | Local + remote |
| Input formats | Pagoda | Seurat | Seurat | Raw | Anndata | raw | Anndata, loom, raw, CellRanger |
| Conversion | ✕ | ✓ | ✕ | ✓ | ✓ | ✓ | ✓ |

(Fig. 1). One HDF5 file or a folder containing multiple such files can then be provided to SCelVis for visualization, and data sets can be selected for exploration on the graphical web interface. To enable both local and cloud access, data can be read from the file system or remote data sources via the standard internet protocols FTP, SFTP, and HTTP(S). SCelVis also provides data access through the open source iRODS protocol (*Rajasekar et al., 2010*) or the widely-used Amazon S3 object storage protocol. The data sources can be given on the command line and as environment variables as is best practice for cloud deployments (*Adam Wiggins, 2011*). The latter allows for easy "serverless" and cloud deployments.

SCelVis is built around two viewpoints on single-cell data (Fig. 1). On the one hand, it provides a cell-based view, where users can browse and investigate cell annotations (e.g., cell type) and cell-specific statistics (sequencing depth, cell type proportions, etc.) in multiple visualizations, e.g., on a t-SNE or UMAP embedding, as violin or box plots or bar charts. Cells to be displayed can be filtered by various criteria, and groups of cells can be defined manually on a scatter plot as input for on-the-fly differential gene expression analysis. On the other hand, SCelVis provides a gene-based view that lets users explore gene expression in multiple visualizations on embeddings or as violin or dot plots. Relevant genes can be specified by hand or selected directly from lists of marker or differential genes.

The source code is available under the permissive MIT license on the GitHub repository at https://github.com/bihealth/scelvis, which also contains a tutorial movie and a link
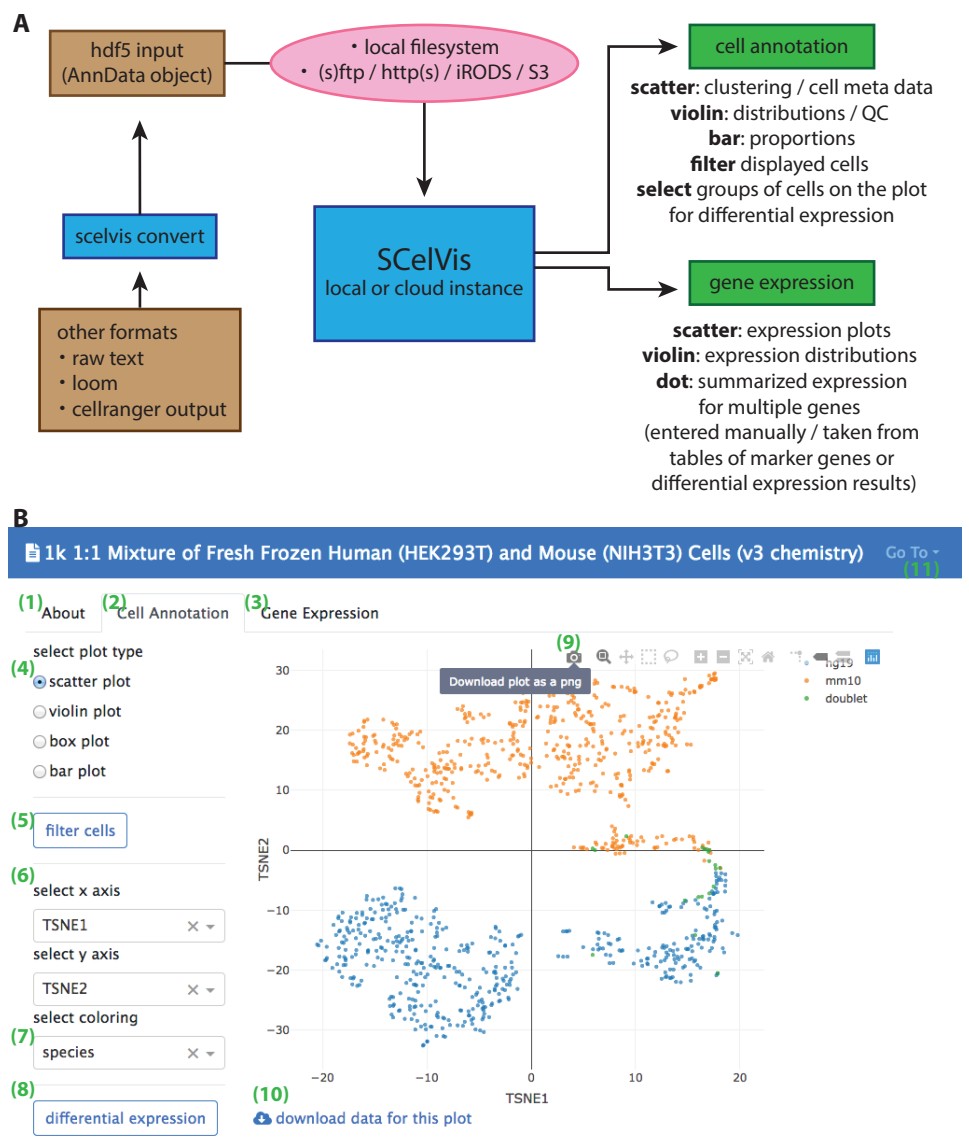

**Figure 1 Overview of SCelVis Architecture and User Interface.** (A) Data can be converted from Cell-Ranger output, loom format or raw text to an input HDF5 file with the SCelVis converter. These files can be uploaded into the web app or loaded remotely via various protocols such as S3, HTTP, etc. SCelVis can then be run locally or on a server/in the cloud and provides various views of the analysis results. (B) Screenshot of the SCelVis interface for a mixture of human and mouse cells from 10X Genomics. Users can browse the "about" tab to obtain background information on the data (1), select the "cell annotation" tab (2) to investigate cell meta data or the "gene expression" tab (3) to interrogate gene expression. The cell annotation view provides scatter, violin, box and bar plots (4). Displayed cells can be filtered (5) by a number of criteria. In typical cases, the scatter plot would be configured with embedding variables on the $x$- and $y$-axis (6) and a categorical or continuous variable for the coloring (7). Differential gene expression (8) can be performed by manually selecting groups of cells on the scatter plot, using "box select" or "lasso select" in hover bar on the top right-hand corner of the plot (9). Here, plot results can also be downloaded in png format. The underlying data can be obtained from a link at the bottom left (10). Other datasets can be selected, uploaded or converted from the menu on the top right (11).

to a public demonstration instance. The software can be run both in the cloud and on workstation desktops via Docker. Documentation and tutorials are provided on https://scelvis.readthedocs.io.

### Usage example

We provide three example datasets within our GitHub repository or via figshare. First, a small synthetic simulated dataset created for testing and illustration purposes, and secondly a publicly available processed scRNA-seq dataset from 10X Genomics containing ∼1,000 cells of a mix of human HEK293T and murine NIH3T3 cells. Finally, we reanalyzed a published data set of stimulated and control peripheral blood mononuclear cells (PBMCs; *Kang et al., 2018*) with the Seurat "data integration" workflow (*Stuart et al., 2019*) and made it accessible via https on figshare (https://files.figshare.com/18037739/pbmc.h5ad; Fig. 2). With the species-mix dataset from 10X, the relevant plot to demonstrate a low doublet rate can be readily re-created (Fig. 2A left; compare to Fig. 2A in *Zheng et al. (2017)*, which shows data obtained with a previous version of the 10X chemistry), and the species composition of the different clusters found by CellRanger can be easily interrogated (Fig. 2A right). For the PBMC dataset, it is straightforward to perform differential gene expression analysis, e.g., between stimulated and control monocytes by using the "filter" and "differential gene expression" buttons (Fig. 2B). Summarized gene expression for cell-type marker genes as well as for general (e.g., IFI6) or cell-type specific (e.g., CXCL10) differential genes can be displayed in a split dot plot as in Fig. 2D of *Stuart et al. (2019)*. Hence, our visualizations for the published datasets are equivalent to those obtained from other visualization tools, e.g., Seurat.

## CONCLUSIONS

In this manuscript, we have presented SCelVis, a method for the interactive visualization of single-cell RNA-seq data. It provides easy-to-use yet flexible means of scRNA-seq data exploration for researchers without computational background. SCelVis takes processed data, e.g., provided by CellRanger or a bioinformatics collaboration partner, as input, and focuses solely on visualization and explorative analysis. Great care has been taken to make the method flexible in usage and deployment. It can be used both on a researcher's desktop with minimal training yet its usage scales up to a cloud deployment. Data can be read from local file systems but also from a variety of remote data sources, e.g., via the widely deployed (S)FTP, S3, and HTTP(S) protocols. This allows for deploying it in a Docker container on "serverless" cloud systems. As both the application and data can be hosted on the network or cloud systems, the application facilitates cross-institutional research. For example, a sequencing or bioinformatics core unit can use it for giving access to non-computational collaboration partners over the internet. This is particularly relevant as it comes with no dependency on any vendor-specific technology such as the Google or Facebook authentication that appears to become pervasive in today's life science.

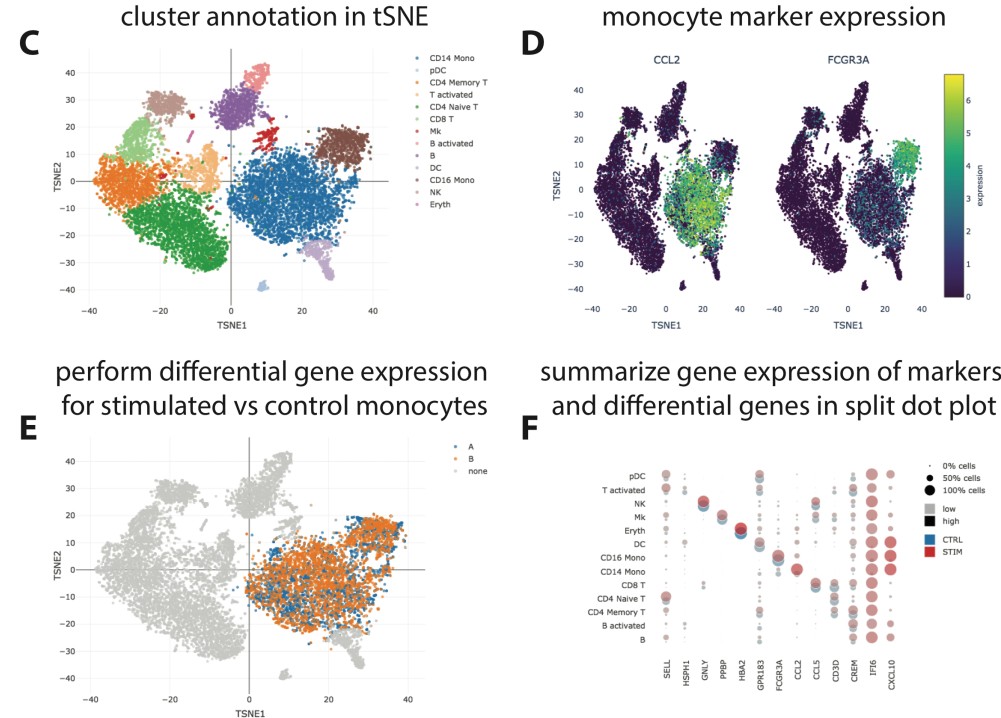

**Figure 2** **Visualization of publicly available scRNA-seq data.** (A + B) scRNA-seq data for a 1:1 mixture of 1k fresh frozen human (HEK293T) and mouse (NIH3T3) cells (Chromium v3 chemistry) were taken from the 10X website (CellRanger output) and visualized with SCelVis. A scatter plot shows human vs. mouse UMI counts per cell and confirms a low doublet rate (A), while a bar plot visualizes the species composition of the different clusters defined by CellRanger (B). (C–F) scRNA-seq data for stimulated vs. control PBMCs (*Kang et al., 2018*). The cluster annotation resulting from the Seurat sample alignment workflow (https://satijalab.org/seurat/v2.4/immune_alignment.html) can be interrogated and monocyte markers can be displayed by selecting from a table of marker genes (C + D). Stimulated or control monocytes can then be isolated using "filter cells" and defined as groups "A" or "B", respectively, for differential expression analysis (E). Summarized gene expression can be displayed for marker genes as well as cell-type specific or globally differential genes in a split dot plot (F).

## ACKNOWLEDGEMENTS

The example dataset for the 1:1 mixture of human and mouse cells processed with CellRanger (v3) was taken from the 10X Genomics website (https://support.10xgenomics.com/single-cell-gene-expression/datasets/3.0.0/hgmm_1k_v3). The example dataset for the stimulated vs. control PBMCs was taken from GEO (accession GSE96583) and re-analyzed with the Seurat sample alignment strategy as explained in the tutorial at https://satijalab.org/seurat/v2.4/immune_alignment.html.

### Funding

The authors received no funding for this work.

### Competing Interests

The authors declare there are no competing interests.

### Author Contributions

- Benedikt Obermayer and Manuel Holtgrewe conceived and designed the experiments, analyzed the data, prepared figures and/or tables, authored or reviewed drafts of the paper, and approved the final draft.
- Mikko Nieminen, Clemens Messerschmidt and Dieter Beule conceived and designed the experiments, authored or reviewed drafts of the paper, and approved the final draft.

### Data Availability

Software is available from GitHub (https://github.com/bihealth/scelvis) and from Zenodo (http://doi.org/10.5281/zenodo.3629501).

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
