# Peer review of "SCelVis: exploratory single cell data analysis on the desktop and in the cloud"

_PeerJ, doi:10.7717/peerj.8607_

## Round 0.1 · original submission · Major Revisions

The manuscript has been carefully evaluated by three reviewers. We find the work very interesting and certainly a useful tool for the community. The manuscript, manual and code need some revisions with especially attention: I) to have more biological focus; ii) documentation and guide more accessible to biologists; iii) to revise the code to a clean one; iv) to consider to expand the functionalities in terms of data visualizaition. We would be happy to consider a revised version which adresses the reviewers' comments.

·

Basic reporting

Writing:
The article has a clear unambiguous and to my knowledge technically correct English with only minor mistakes that can easily be corrected:

l. 56 “communication need” -> “communication needs”
l. 68 “Further, ,” -> “Further,”
l. 68 “(yet)” -> "yet" or "remove"
l. 73 “… Docker image strongly depend on” -> “… Docker image strongly depends on”
l. 84 “graphic web” -> “graphical web”
l. 129 “in more complex framework” -> “in a more complex framework” / “in more complex frameworks”


References:
The article contains an appropriately amount of references to back up the content and all references matches the content they have been placed in.
A DOI seem to be missing at:
l. 158-160. You might add this: https://doi.org/10.1038/sdata.2016.18


Sections:
Overall I like the different sections. However, the discussion part seems to be very vague and more resembles a summary of the paper rather than a discussion. In that sense it resembles a slightly more detailed version of the conclusion part. If both a discussion and a conclusion are necessary to be included I suggest that the authors in the discussion elaborate a little about for instance the perspectives in the field of single cell analysis/visualization, what the next steps might be (either for the tool or the community in general) to improve these kind of analysis/visualization tools.


Figures:
I generally like the figures in this paper but I have some comments on them:

Figure 1: I think it is a bit of a pity that the figures are so small that you cannot see the details of them so easily. For instance, it is not possible to see the legends on the gene-centric plots. If not making the plots bigger a possible solution could be to make them vectorized so that the reader can zoom in and read. With the figures I received, this is not possible due to the resolution. Also, I would also suggest to briefly describe the two visualization fields “cell-centric view” and “gene-centric view” in the figure caption.

Figure 2: It seems a bit odd that the plots are overlapping, especially when there are only two plots. Again, I think it would be better with higher resolution/vectorized plots. With the current settings it quickly becomes pixilated when zooming in.


Raw code on GitHub:
Generally well-documented code and this I think don't need further improvement to pass. FYI: There are some TODO’s in the code. Also, small inconsistencies are observed in the documentation style in for instance in _version.py. For instance the multiline documentation style for describing the function is a bit different between the Get_keywords & versions_from_parentdir functions. There also seem to be differences in the ways the functions are described. Some functions have formal sentences, while other have first/third person comments e.g. ‘I’/’you’/’we’. I think the documentation could be improved by using a more consistent style instead of mixing the two styles.

Experimental design

no comment

Validity of the findings

no comment

Reviewer 2 ·

Basic reporting

The paper did not report the advantage of this tool among other available tools.

Experimental design

The authors in this paper developed a web tool to improve the analysis of single-cell RNA data for non-bioinformatician.

Validity of the findings

I run the tool but gets error after uploading the data

Additional comments

The authors in this paper developed a web tool to improve the analysis of single-cell RNA data for non-bioinformatician. The tool is very beneficial to the field however many limitations should be addressed.
1- I had a difficult time running the tool because of unclear guidance and lack of tutorial. I strongly recommended making a Youtube video for users with all details.
2- Made some examples of Tags such as the below two lines(not everyone understand tags):
docker pull quay.io/biocontainers/scelvis:0.5.0--py_1
docker run -p 8050:8050 -v data:/data quay.io/biocontainers/scelvis:0.5.0--py_1 scelvis run --data-source /data
docker run quay.io/biocontainers/scelvis:0.5.0--py_1 scelvis --help

3- http://localhost:8050 worked for me not http://0.0.0.0:8050/.
4- When I uploaded the data: hgmm_1k.h5ad, I got nothing.
5- When I converted the data "dummy_raw.zip", I got the below error:\Internal Server ErrorThe server encountered an internal error and was unable to complete your request. Either the server is overloaded or there is an error in the application.
6- Scatter, violin and dot plots are not sufficient for the tool.
7- I recommend authors to look at other available tools:https://amp.pharm.mssm.edu/biojupies/
8- Authors should compare existence tools with their tool. What is the advantage?

·

Basic reporting

The paper “SCelVis: Powerful explorative single cell data analysis on the desktop and in the cloud” describes the software for visualization of single-cell data with the ability to run both on the desktop and in the cloud. The paper nicely explains the background by the development of this software and shows examples of the software when used.

In general I think the text is nicely written, however, it focuses a lot more on the computational advantages of the method, than on the biological part. If the package is aimed towards biologists with close to no bioinformatics background, I think this should to some extent be reflected in the language of the paper. I do not think the introduction to the computational advantages should be removed from the paper, but some more focus on why it is advantageous to use the software as a biologist would improve the paper considerably.

The language could be improved the following places:
- Line 40-42: This sentence seems a bit out of place compared to the rest of the paragraph.
- Line 50-56: I think this explanation is too long and ends up overcomplicating the aim of the package.
- Line 61-76: The walkthrough of current tools is really informative, but the flow of this paragraph could be improved.
- Line 90: This sentence is unclear and maybe unnecessary.

The language could be less subjective in the following places:
- Line 63: “to our knowledge quite rare” is a vague statement.
- Line 114-115: the sentence seems subjective and unclear.
- Line 122: The use of “interesting” is subjective.
- Line 126 + paper title: the use of the word powerful seems subjective and not sufficiently supported by the results in the paper

There are minor mistakes the following places:
- Line 68: two commas
- Line 73: depend -> depends
- Line 91: On the one hand -> On one hand
- Line 105: 1000 is the exact number of cells in the dataset, the tilde should be removed
- Line 173: 10x website -> 10x Genomics website

I would suggest the authors to make the paragraph on the different options of data visualization more specific, e.g. instead of “such as cell type” on line 92, it should state exactly the annotations that are possible to view here. If this depends on the annotations provided in the data preprocessing, this should be mentioned. And the phrasing “such as” should be changed.

Figure 1:
- From the figure now it looks like both an hdf5 file and raw text / cellranger output is needed to run SCelVis. The figure would be better if constructed in a way that makes it clear that only one of these data sources is required. This should also be underlined in the figure text.
- The axes and letters of the plots are too small. Either the authors should increase the font size so that the numbers and letters are readable, or the plots could be simplified by removing numbers and letters, if the data presented in this figure is not used for further analysis.
- The plots in general are too small to be clearly understandable, and they could be simplified by e.g. focusing on fewer samples and/or genes.
- The colors of the boxes are relatively dark on print. Additionally, there’s a lot of colors on the figure, which does not facilitate the understanding. I would suggest the authors to also simplify the plot color-wise.

Figure 2:
- The overlay of A and B does not work properly, as it hides the majority of the plot in A.
- The text in the screenshots is too small to be easily read.

Experimental design

I think the paper needs a more detailed research question, and a closer link to the biological research that this package facilitates. I would strongly suggest the authors to improve this through more analysis and demonstration of findings that are possible with this package, for instance by focusing more on the results presented in the figures and commenting on these results in the text. The use of the publicly available dataset from 10x Genomics is fine for this.

I would like a bit more background on what a AnnData object is, why this format was chosen, and especially which data to convert into AnnData – in line 80 it says raw text, but it is unclear whether this is in the form of raw counts or log-normalized data. Additionally, the option to upload raw text is non-existing in the user interface, only the option to upload and convert CellRanger Output is found.

I would suggest the authors to improve the discussion by including more perspective on why to use this package in addition to “visualization and explorative analysis” on line 114. There should be some suggestions to the nature of this explorative analysis. Additionally, I would suggest the authors to do a more direct comparison to some of the methods described in the introduction.

I think a simulated dataset is superfluous here, as it can only be used for illustration purposes, a purpose that the provided dataset from 10x Genomics also fulfills.

Validity of the findings

The example datasets are not provided in the package, as the paper states, only found on the Github.

Additional comments

I think the software presented in this paper could be of value to some researchers working with single-cell RNA-sequencing data. However, to best present the advantages of this software, I think some points in the paper needs revision before publication, especially describing the analyses that are possible to conduct using this software.

Minor comments for improvement:
- The option to add logarithmic axes on the violin and scatter plots could be beneficial in some analyses.
- I think other words than “perspective” in line 91 could better reflect the design of the software, for instance “viewpoints”.

---

## Round 0.2 · Major Revisions

One of the reviewers still has important criticisms which need to be addressed especially with attention to: I) comparison with other tools; ii) speed of the code; iii) clarity for biologists in the documentation; iv) more visualization support. We would be glad to reconsider a revised version addressing these points.

·

Basic reporting

no comment

Experimental design

no comment

Validity of the findings

no comment

Reviewer 2 ·

Basic reporting

scelvis is cloud tool to analyze single cell RNA. However, there is no clear statement on how scelvis is different from the existing tools.

Experimental design

I encourage the author to compare his tool with other tools such as the below table
https://www.ncbi.nlm.nih.gov/pmc/articles/PMC5716224/table/Tab1/?report=objectonly
Users need to know what new features scelvis can give them.

Validity of the findings

My major concern about scelvis is the novelty. It only provides violin and scatter plot for the data. The tool is not easy to be run by biologist without bioinformatics background. It also needs long time to upload even small data (I waited 15 min to upload a data with size 50 M and still could not upload the data). There is a limitation of this tool as it only provide violin plot and scatter plot. I encourage authors to improve its GUI, add more features and solve the speed problem.

Additional comments

The tool is not easy to be run by biologist without bioinformatics background. It also needs long time to upload even small data (I waited 15 min to upload a data with size 50 M and still could not upload the data). There is a limitation of this tool as it only provide violin plot and scatter plot. I encourage authors to improve its GUI, add more features and solve the speed problem.

---

## Round 0.3 · accepted · Accept

I am glad to endorse your manuscript for publication, all the remaining comments by the reviewer have been nicely addressed.

Reviewer 2 ·

Basic reporting

A lot of improvement have been done.

Experimental design

Authors added table 1 that compared their algorithm with available packages. They also improved the tutorial and the documentation.

Validity of the findings

I recommended if they add a paragraph about the limitations of the tool and what should be done in the future to expand its usage.